# Identification of age-specific gene regulators of La Crosse virus neuroinvasion and pathogenesis

Rahul Basu[1,2], Sundar Ganesan[3], Clayton W. Winkler[1], Sarah L. Anzick[4], Craig Martens[4], Karin E. Peterson[1,5] ✉ & Iain D. C. Fraser ●[2,5] ✉

One of the key events in viral encephalitis is the ability of virus to enter the central nervous system (CNS). Several encephalitic viruses, including La Crosse Virus (LACV), primarily induce encephalitis in children, but not adults. This phenomenon is also observed in LACV mouse models, where the virus gains access to the CNS of weanling animals through vascular leakage of brain microvessels, likely through brain capillary endothelial cells (BCECs). To examine age and region-specific regulatory factors of vascular leakage, we used genome-wide transcriptomics and targeted siRNA screening to identify genes whose suppression affected viral pathogenesis in BCECs. Further analysis of two of these gene products, Connexin43 (Cx43/*Gja1*) and EphrinA2 (*Efna2*), showed a substantial effect on LACV pathogenesis. Induction of Cx43 by 4-phenylbutyric acid (4-PBA) inhibited neurological disease in weanling mice, while *Efna2* deficiency increased disease in adult mice. Thus, we show that *Efna2* and Cx43 expressed by BCECs are key mediators of LACV-induced neuroinvasion and neurological disease.

La Crosse Virus (LACV), a negative-sense RNA virus belonging to the bunyaviridae family[1], is one of the leading causes of arboviral encephalitis in children[2]. In adults, LACV infection generally causes a very mild, febrile syndrome[3]. The higher incidence of neurological disease in children compared to adults suggests age-related differences that may be responsible for the ability of the virus to gain access to the central nervous system (CNS) and/or cause damage within the CNS. Understanding these differences could provide avenues for prevention or treatment of LACV encephalitis.

The blood-brain barrier (BBB) is a selectively permeable barrier that plays an important role in inhibiting the access of pathogens to the CNS. The blood-brain barrier (BBB) is comprised mainly of brain capillary endothelial cells (BCECs) with neighboring astrocytes, basement membrane and pericytes, which interact with surrounding neurons and microglia to form the neurovascular unit[4]. The selective permeability across the BCECs is defined by several transporter and carrier proteins along with the tight junction (TJ), adherens junction (AJ) and gap junction (GJ) (collectively cell junction (CJ)) proteins[5,6]. The integrity of the BBB is strengthened during development[7,8]. It reaches its peak in adulthood and then gradually weakens with age due to reduced CJ protein expression and functional impairment of transporters[9,10].

The pathology observed from LACV encephalitis indicates substantial breakdown of the BBB. Immunohistochemistry (IHC) analysis of LACV patient brain biopsies shows perivascular mononuclear cuffing with focal aggregates of immune cells[11], and further studies have

[1]Neuroimmunology Section, Laboratory of Persistent Viral Disease, Rocky Mountain Laboratories, NIAID, NIH, 903 S. 4th StreetMT 59840 Hamilton, USA. [2]Signaling Systems Section, Laboratory of Immune System Biology, National Institute of Allergy and Infectious Diseases, National Institutes of Health, 4 Memorial Drive, Bethesda, MD 20892, USA. [3]Research Technologies Branch, National Institute of Allergy and Infectious Diseases, National Institutes of Health, 4 Memorial Drive, Bethesda, MD 20892, USA. [4]Genomics Research Section, Research Technologies Branch, Division of Intramural Research, National Institute of Allergy and Infectious Diseases, National Institutes of Health, 903 S. 4th Street, MT 59840 Hamilton, MT, USA. [5]These authors contributed equally: Karin E. Peterson, Iain D. C. Fraser. ✉e-mail: petersonka@niaid.nih.gov; fraseri@niaid.nih.gov

demonstrated LACV encephalitis-induced vascular injury[12]. Thus, LACV encephalitis is closely associated with neurovascular pathology and abnormalities.

A similarly age dependent LACV encephalitis and vascular pathology is observed in the C57BL/6 mouse model. Weanling mice (~3 weeks of age) are highly susceptible to LACV infection of the CNS and develop clinical disease following either peripheral (intraperitoneal, IP) or direct CNS infection (intracerebral, IC). In contrast, adult mice ≥6 weeks of age are resistant to peripheral infection (IP) but are susceptible to direct CNS (IC) infection[13–15]. Thus, there is a clear age-related difference in the ability of LACV to gain access to the CNS following peripheral infection.

Studies on age-related susceptibility show increased vascular leakage and breakdown of the BBB in young mice following LACV infection. Although LACV infected BCECs are not readily observed in vivo, the breakdown of the BBB is mediated by leakage specifically through the microvessels/BCECs in the olfactory bulb (OB)/ anterior olfactory nucleus (AON) region, but not those in the cortex (CT) region[16]. The viral leakage generally peaks at 3 days post-infection (dpi) and viral infection of neurons is observed in regions associated with this vascular leakage. In a previous study we have demonstrated that an age-specific response of brain microvessels/ BCECs causes differential cytopathic effects and LACV susceptibility. Additionally, using ex vivo isolated brain microvessel fragments and in vitro cultures of primary BCECs isolated from weanling and adult mice, we showed that weanling BCECs are more prone to LACV infection, formation of syncytia-like aggregates and bystander cell death[17]. BCEC responses may be controlled by several factors including the host immune response, cell survival, expression of functional CJ proteins, maintenance of blood vessels or by multigenic interactions. Identifying factors that differ between weanling and adult brain microvessels/ BCECs during LACV infection may provide insights into the factors that are critical in preventing LACV neuroinvasion.

We hypothesized that putative restriction factors may be present or enhanced in the LACV-resistant adult mice, while putative susceptibility factors may predominate in the LACV-susceptible weanling mice. To identify such factors, we first used RNA sequencing (RNA-seq) based transcriptome analysis in ex vivo isolated brain microvessel fragments obtained from weanling and adult mice and assessed responses to polyinosinic: polycytidylic acid (poly I:C) to mimic virus-induced immunostimulation. We further compared RNA-seq profiles in brain microvessel fragments isolated from the OB and CT region of LACV-infected or mock-inoculated weanling mice. Candidate genes showing differential expression profiles from these analyses were subjected to a targeted small interfering RNA (siRNA) screen to identify putative regulators of age dependent LACV susceptibility. Hit genes were evaluated for their site of potential involvement in the LACV-infectious pathway, and two genes, gap junction protein alpha 1 (*Gja1*) and ephrinA2 (*Efna2*) were identified as potential therapeutic targets to control LACV-induced vascular leakage and establishment of neuroinfection.

## Results

### RNA-seq analyses of adult and weanling brain microvessels identify candidate gene regulators of LACV infection

We first sought to identify candidate gene regulators that contribute to age-dependent susceptibility to LACV. Weanling (W) and adult (A) mice were treated with either poly I:C (PI) for 3 h (as a surrogate for a viral RNA stimulus) or vehicle control (V). Whole brain microvessel fragments isolated from these mice were subjected to RNA-seq analysis to identify both intrinsically age dependent or Poly I:C stimulation induced genes (Fig. 1a). Gene expression was compared for basal level age-related differences comparing weanling vehicle (WV) versus adult vehicle (AV) groups. Activation comparisons were done by comparing Weanling poly I:C stimulated (WPI) versus WV, Adult poly I:C

stimulated (API) versus AV as well as WPI versus API. Differential gene expression volcano plots and pathway enrichment analyses (Ingenuity Pathway Analysis, IPA), from the above comparisons highlight the cellular processes that distinguish the different treatment and age groups (Supplementary Fig. 1a–d). Specific genes for further evaluation were selected based on differential expression between groups with a log2 difference of ≥1, $p < 0.05$ and base-mean ≥100 (Supplementary Table 1, Group 1). To evaluate the likely cell composition in the isolated microvessel fragments, we assessed expression of cell type-specific genes from the RNA seq data (Supplementary Table 2). This showed that BCEC-specific genes were highly enriched, there was moderate/low expression of pericyte markers, but very little expression of genes characteristic of astrocytes, neurons, and smooth muscle cells. This confirmed that the method used for microvessel fragment preparation resulted in highly specific BCEC enrichment, as described previously[18–20].

As regional differences in BBB integrity are observed following LACV infection and virus infection is first observed in the OB region in weanling mice[16], we also completed an RNA-seq analysis on microvessel fragments isolated from different regions of LACV-infected (L) and mock-inoculated (M) weanling mouse brains at 3 dpi. Genes from this comparison were selected based on two differential gene expression criteria: either a difference in expression between microvessel fragments from the OB of LACV-infected mice (LOB) versus the OB of mock-infected mice (MOB), or a difference in gene expression between LOB versus microvessel fragments from CT of LACV-infected mice (LCT) (Fig. 1b). These criteria were used to select additional genes for further analysis (Supplementary Table 1, Group 2). Differential gene expression and IPA analyses from these comparisons highlight the expected enrichment of interferon and neuroinflammation following LACV infection of the OB, and differences in Gap Junction and Ephrin signaling in the OB/CT comparison (Supplementary Fig. 1e, f). Finally, in addition to the selected Group 1 and 2 genes, we also included genes known to affect BBB function and the host immune response to LACV, but not necessarily differentiated in the above RNA seq analysis (Supplementary Table 1, Groups 3 and 4, respectively).

To further confirm expression differences of the selected genes, we analyzed expression by qPCR following LACV infection. RNAs were extracted from microvessel fragments obtained from LACV-infected adult (AL) or weanling (WL) mice at 3 dpi as well as mock-inoculated weanling (WM) and adult (AM) mice, while additional microvessel fragments were extracted from OB and CT regions of LACV infected (3 dpi) weanling and adult mice. Real-time PCR analyses on ex vivo isolated microvessel fragments showed genes falling into four main categories, adult-enhanced, weanling enhanced, LACV-induced and region specific (OB-CT) (Fig. 1c–f). For example, ephrinA2 (*Efna2*; an angiogenic molecule) and *H2q6* (belonging to MHC class I family) were expressed at higher levels in adult microvessel fragments, regardless of virus infection (Fig. 1c). Bone marrow stromal cell antigen 1 (*Bst1*; an immune regulator) and matrix metallopeptidase 25 (*Mmp25*) had higher basal levels in weanling mice (Fig. 1d). C-type lectin domain family 4, member e (*Clec4e*; a molecule that recognizes LACV and plays a limited role in early antiviral responses against LACV[21]) and claudin 1 (*Cldn1*; a TJ protein present in endothelial/epithelial cells) were increased by LACV infection in adult mice, but not in weanling mice (Fig. 1e). Only a few genes were differentially expressed between OB and CT microvessel fragments. Among these, two homeostasis maintenance genes aquaporin 1 (*Aqp1*; a passive transporter) and transthyretin (*Ttr*; a carrier protein) were increased in CT microvessel fragments from both weanling (WCT) and adult (ACT) mice compared to OB microvessel fragments obtained from weanling (WOB) and adult (AOB) mice. The pairwise comparison between the OB and CT microvessel fragments are shown for each age group (Fig. 1f). Thus, we identified genes by RNA seq (Fig. 1a, b) and qPCR analyses (Fig. 1c–f)

that were differently expressed in microvessel fragments by age, infection, or region for further study.

## Targeted siRNA screen elucidates potential LACV susceptibility or resistance factors in endothelial cells

To directly examine whether the selected genes may influence viral pathogenesis, we utilized an in vitro model of endothelial cell LACV infection. We previously found that, although infected endothelial cells are not readily detectable in vivo, they are infected in an age-dependent manner in vitro and this infection is associated with direct and bystander damage to endothelial cells[17]. To establish a gene-perturbation cell-based screening assay, we chose a mouse endothelioma cell line, bEnd.3, as these cells could be infected with LACV and were readily amenable to siRNA transfection. To optimize siRNA delivery, we employed lethal and nontargeting (NT) control siRNAs to test different transfection reagents and identify conditions with

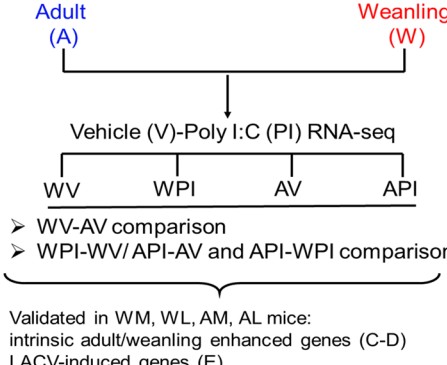

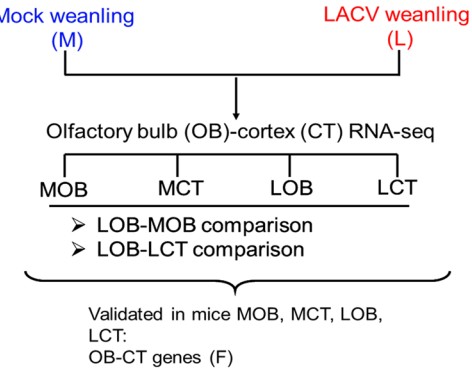

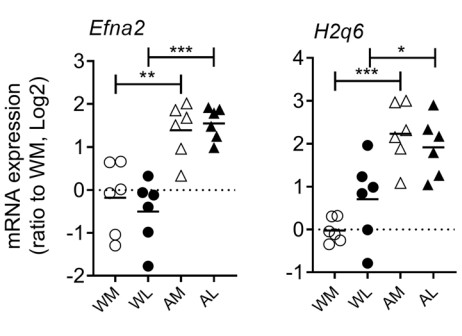

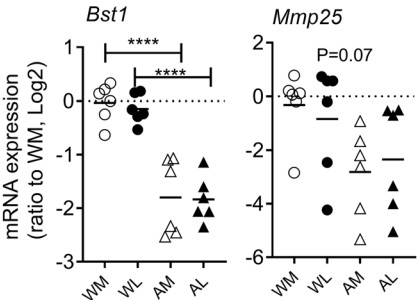

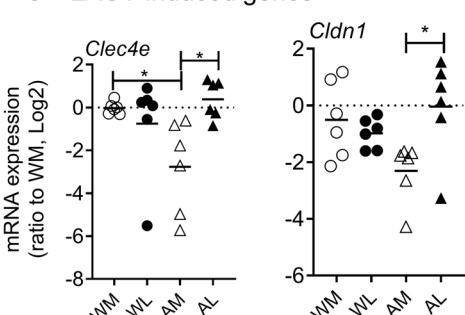

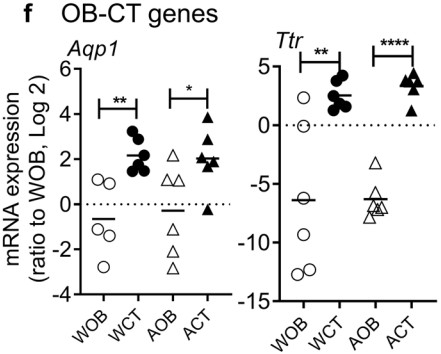

**Fig. 1 | RNA-seq analyses and validation of target genes ex vivo. a** Sequential approach to select target genes from RNA-seq analyses on vehicle (V) or Poly I:C (PI)-treated weanling (W) or adult (A) mice (WV, WPI, AV, and API). **b** Identification of target genes from RNA-seq analyses of microvessel fragments from olfactory bulb (OB) and cortex (CT) regions of weanling LACV infected (LOB and LCT) and mock inoculated mice (MOB and MCT) (N = 3 for all RNA-seq samples). The default DESeq function with betaPrior = FALSE was applied, wherein a negative binomial generalized model with Wald test for significance was performed. The two-sided p-values were corrected for multiple testing using the Benjamini Hochberg method. **c–e** Differential expression of representative genes, as validated by real-time PCR analyses on ex vivo isolated microvessel fragments from weanling mock (WM), weanling LACV (WL), adult mock (AM) and adult LACV (AL) infected mice. Gene categories: (**c**) intrinsic adult-specific (for *Efna2* and *H2q6* WM vs AM P = 0.0032 and P = 0.0001 and WL vs AL P = 0.0002 and P = 0.0356), **d** intrinsic weanling-specific

(for *Bst1* and *Mmp25* WM vs AM P < 0.0001 and P = 0.1010 and WL vs AL P < 0.0001 and P = 0.4684) or (**e**) LACV-infection induced, adult enhanced genes (for *Clec4e* and *Cldn1* WM vs AM P = 0.0453 and P = 0.0894 and WL vs AL 0.6490 and P = 0.5686). **f** Differential expression of representative genes from OB and CT microvessel fragments obtained from LACV-infected adults (AOB and ACT) and weanlings (WOB and WCT). P = 0.0043 and P = 0.0068 for *Aqp1* and *Ttr* for WOB vs WCT comparison and P = 0.0417 and P << 0.0001 for *Aqp1* and *Ttr* for AOB vs ACT comparison. The data in (**c–f**) were transformed to log2 scale for a more normal distribution and the datapoints were statistically analyzed or plotted post-transformation. Significance values were measured by one-way ANOVA followed by Tukey's multiple comparison (**c–e**) or two-tailed, multiple unpaired t-tests (**f**) (N = 5–6 for all mouse experiments, from Panel **c–f** and individual datapoints are shown). (\*P < 0.05, \*\*P < 0.01, \*\*\*P < 0.001 and \*\*\*\*P < 0.0001).

maximum cell killing with lethal siRNA while maintaining maximum viability with the si-NT control. This identified an optimal delivery protocol using 50 nM siRNA with the Transit TKO transfection reagent (Supplementary Fig. 2a). We then established three control conditions for the screen: a) a baseline condition for LACV infection with NT siRNAs, b) a control for known restriction factors using a combination of siRNA to *Ifnar1* and *Ifnar2* (si-*Ifnar* or si-Upreg. Control; upregulated LACV infection) and c) a control for known susceptibility factors using a combination of siRNAs to Clathrin heavy chain and Rab5a (si-*CltcRab* or si-Downreg. Control; downregulated LACV infection) (Supplementary Fig. 2b, c).

We then screened the 35 selected genes shown in Supplementary Table. 1 along with the three control conditions at three different phases of LACV infection on BCEC monolayers at room temperature. The first was the early phase (6 hpi; Fig. 2a), which would suggest an effect on virus entry. The second was the medial infectious phase (24 hpi; Fig. 2b), suggesting an effect on virus replication and spread through the endothelial cell culture. These kinetic phases have been similarly observed for LACV infection in a different cell line[22]. The cell-based assay for these first two phases used an anti-LACV antibody to measure viral fluorescence intensity on a per-cell basis. This was determined by measuring LACV sum-intensity (which is impacted by both the number of infected cells as well as the intensity of infection) and then quantitating the ratio to Hoechst area (which normalizes the intensity value to total cell number). The last, late phase assay (72 hpi; Fig. 2c) measured cell loss through LACV-induced cell death. Using these criteria, knockdown of putative restriction factors would be expected to give increased LACV intensity and reduced cell viability, while knockdown of putative susceptibility factors would lead to reduced LACV intensity and higher cell viability. We identified two clusters of genes that showed putative resistance or susceptibility phenotypes (color highlighted in Fig. 2a−c), with representative images of virus upregulation or downregulation shown in Supplementary Fig. 3.

Putative restriction factors identified in the primary screen included *Efna2*, *Clec4e*, *Gja1* (expressed as Connexin43 protein, abbreviated as Cx43), interferon induced transmembrane protein 3 (*Ifitm3*, an RNA viral restriction factor), lymphocyte antigen 6 complex locus C2 (*Ly6c2*) and *H2q6*. For *H2q6*, there was a variation between 6 and 24 hrs in phenotype, with increased viral intensity only observed at the early time point (Fig. 2a, b). Clusters of syncytia-like aggregation and multinucleated cells were observed in *H2q6* knockdown cultures, which might explain the limitation of the increased viral intensity phenotype to the early time point (white arrows, Supplementary Fig. 3a, b). *Gja1* knockdown caused dissociation of the cellular monolayer (white arrow, Supplementary Fig. 3b), consistent with its established role in gap junction integrity. Putative susceptibility factors included vascular endothelial growth factor A (*Vegfa*), actin-cytoskeleton remodeling protein gelsolin (*Gsn*), claudin 2 (*Cldn2*), claudin 5 (*Cldn5*) and tight junction protein 2 (*Tjp2/ZO2*) (Fig. 2a−c). Thus, multiple genes were identified that either enhanced or reduced LACV in vitro infection of bEnd.3 endothelial cells.

**Putative target genes control susceptibility of endothelial cells to LACV infection in different phases of viral infection**
Based on the targeted siRNA screen, we chose seven putative restriction factors (Group1 genes: *Bst1*, *Efna2*, *H2q6*, *Clec4e*, *Gja1*, *Ifitm3* and *Ly6c2*) for further validation in primary adult and weanling BCECs. Interestingly, knockdown of all of these genes increased susceptibility of adult primary BCECs to LACV infection (Fig. 3a), while only *Ifitm3* knockdown showed a significantly increased infection level in primary weanling BCECs (Fig. 3b), supporting the premise that these genes may contribute to the increased resistance to LACV in adult BCECs.

We also tested putative susceptibility factors for validation in primary culture (Group2 genes: *Cldn2*, *Tjp2*, *Cldn5*, *Vegfa*, *Mmp8*,

*Mmp25* and *Gsn*). Knockdown of *Mmp25*, a weanling enhanced gene, led to reduced LACV intensity in both adult and weanling primary BCECs (Fig. S4), suggesting this host gene may facilitate LACV infection. However, only marginal effects were observed for the other putative susceptibility factors in primary BCECs. Since we observed a higher frequency of hit validation for Group1 genes in primary cells, we focused on genes from this group for further analysis.

In an effort to dissect mechanisms of virus inhibition, we next utilized the bEnd.3 cells to examine the stages at which perturbation of the putative restriction factors may limit virus replication. Attachment / entry assays (inducing viral binding at low temperature, see methods section) showed that two genes, *Ly6c2* and *Ifitm3*, restricted viral entry (Fig. 3c). Compared to si-*Ly6c2*, si-*Ifitm3* knockdown caused smaller upregulation of infected cell count as well as LACV intensity/cell (Fig. 3c, d). We then conducted plaque assays and found that perturbation of *Ly6c2*, *Gja1* and *Efna2* increased virus production at 24 hpi (Fig. 3e).

In our primary screen, we had noted that knockdown of the *H2q6* gene also induced syncytia-like aggregation formation in bEnd.3 cells (Supplementary Fig. 3a, b). To investigate this further, control and *H2q6*-perturbed bEnd.3 cells were co-stained for both LACV and F-actin. It has been reported previously that actin cytoskeletal proteins are altered in OB BCECs following LACV infection[16]. We observed that LACV colocalized with the actin network in control cells, specifically at the cellular periphery and lamellipodial extensions (thin arrow, upper panel, Fig. 3f). In the *H2q6* knockdown condition however, specifically in the cells having two or more nuclei (forming syncytia-like aggregation), we observed disruption of regular, filamentous actin staining and granular staining of actin and LACV located around cellular nuclei (thick arrow, bottom panel, Fig. 3f) suggesting a potential alteration of virus trafficking. Thus, in vitro validation of the putative viral restriction factors (*Efna2*, *H2q6*, *Clec4e*, *Gja1*, *Ifitm3*) identified from our focused siRNA screen suggests possible roles for select host genes in restricting or altering viral entry, trafficking, or viral particle release in LACV-infected endothelial cells.

**Efna2 as a regulator of LACV susceptibility in vivo**
One of the primary genes that significantly affected LACV infection of endothelial cells was *Efna2*, which is intrinsically abundant in adult BCECs. The encoded EFNA2 protein is an angiogenic factor[23] that carries out signaling responses via EphrinA (EphA) class receptors[24]. To further examine EFNA2, we tested whether recombinant mouse EFNA2 (rec-EFNA2) protein could restrict LACV infection in weanling and adult primary BCECs. Weanling and adult primary BCECs were cultured, infected with LACV (10 multiplicities of infection; abbreviated as MOI), then maintained in EFNA2 conditioned or control medium for up to 72 h post infection. Two different EFNA2 concentrations, low (2ug/ml) or high (20 ug/ml) were used. Upon rec-EFNA2 treatment, weanling BCECs showed reduced levels of LACV infection at 24 hpi, which was significant for the high concentration of EFNA2 (Fig. 4a). The adult BCECs, already mostly resistant to LACV infection, showed partial but not significant reduction in infection. Thus, exogenously added EFNA2 can restrict in vitro LACV replication in weanling BCECs (Fig. 4a).

To examine whether *Efna2* has a role in restricting LACV infection of the CNS in vivo, we analyzed the development of neurological disease in normally resistant adult mice deficient in *Efna* family members. We initially tested >6weeks old mice that had mixed deficiency genotypes for *Efna2*, *Efna3* and *Efna5* (*Efna2*[−/−] mixed; abbreviated as *Efna2*[−/−] (m)), as detailed in Supplementary Table 3. All mice were distributed in two groups: homozygous for *Efna2* deficiency with mixed [+/+], [+/−] or [−/−] genotypes for *Efna3* and *Efna5* (*Efna2*[−/−] (m) or heterozygous for *Efna2* deficiency with mixed [+/+], [+/−] or [−/−] genotypes for *Efna3* and *Efna5* (*Efna2*[+/−] (m)) as well as wildtype mice with *Efna2*[+/+]3[+/+] 5[+/+] genotype (wildtype, abbreviated as WT). Following inoculation of 10⁵ PFU/mouse, approximately 50% of *Efna2*[−/−] (m) mice developed

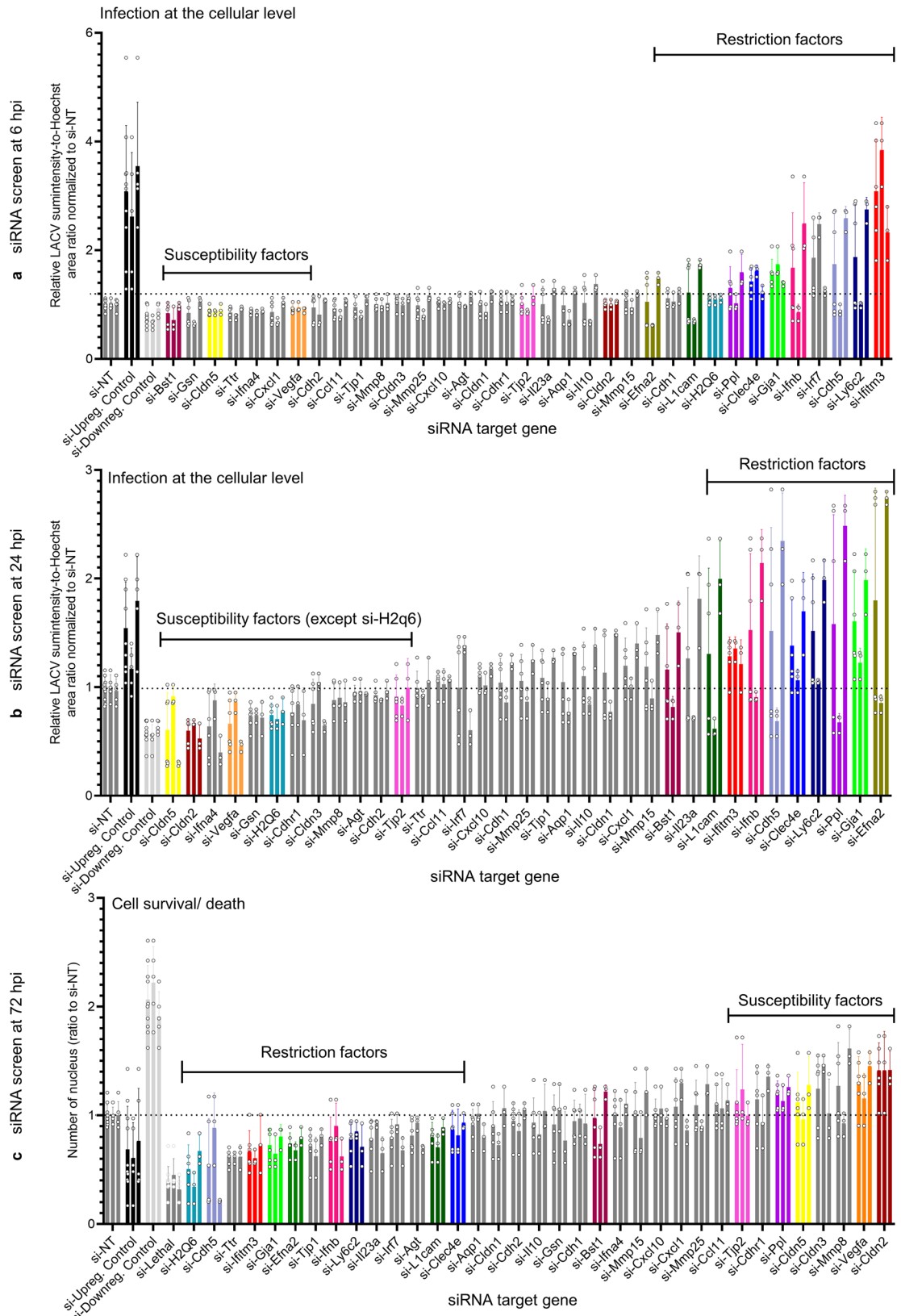

**Fig. 2 | Identification of putative host restriction and susceptibility factors in LACV infection through targeted siRNA screening.** bEnd.3 cells were transfected with 50 nM siRNA against the indicated target genes (for 72 h), along with viral upregulation control (i.e., si-*Ifnar1* and *Ifnar2*, abbreviated as si-Upreg. Control), downregulation control (i.e., si-*Cltc* and *RabSa*, abbreviated as si-Downreg. Control), non-targeting (si-NT, dotted line) and cell death (si-Lethal) controls where indicated. **a**, **b** Analyses of degree of infection (10 MOI of LACV) normalized to si-NT (LACV sum-intensity-to-Hoechst area ratio) at 6 (**a**) and 24 hpi (**b**). **c** Analyses of cell survival (nuclear count) at 72 hpi. The first bar represents the average of 2 independent gene-specific siRNA, whereas the next two bars represent siRNA#1 and #2 for each gene (mean ± SD, *N* = 3 each individual siRNA). Putative hit genes are color-matched across the different graphs.

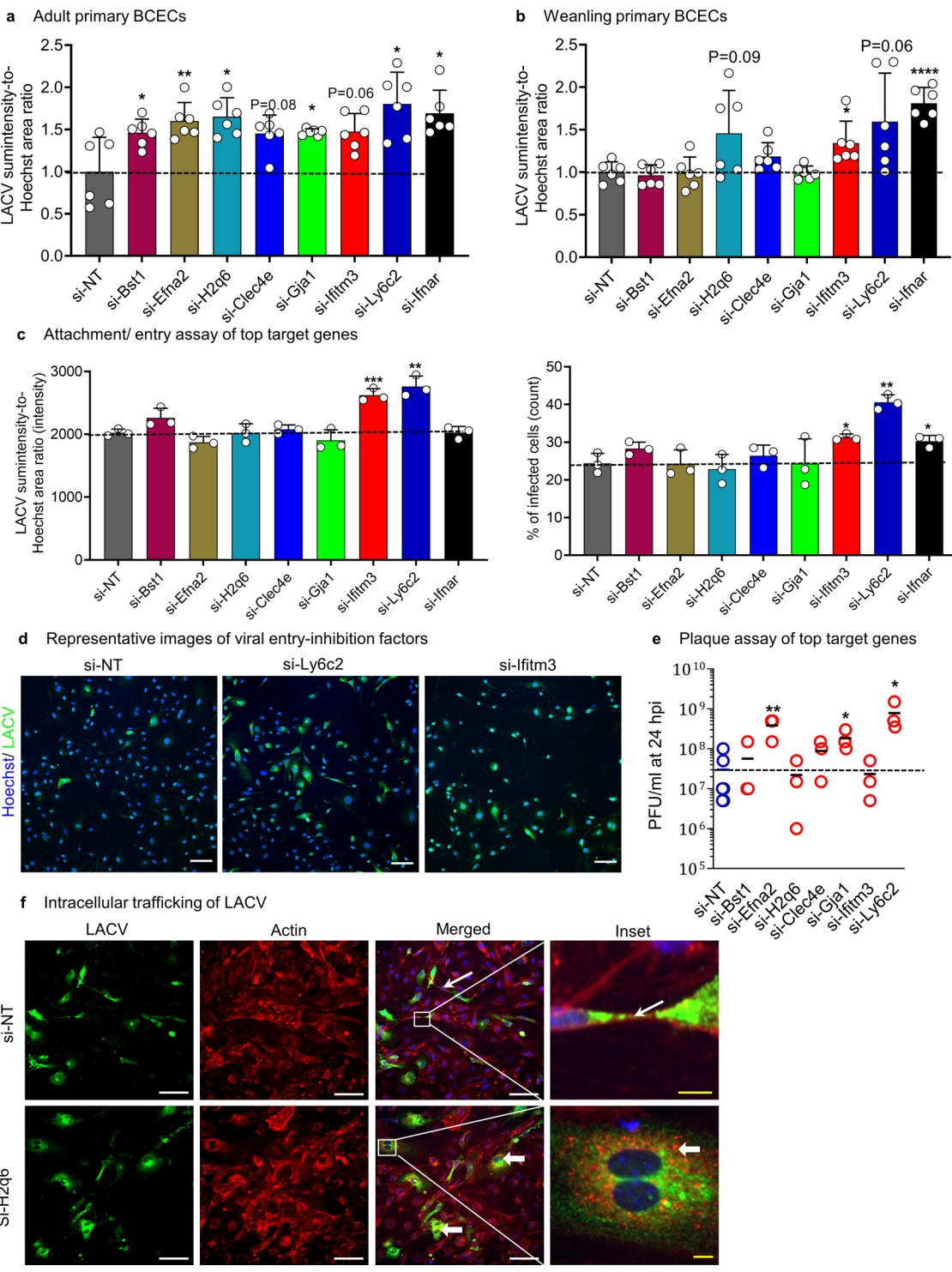

**Fig. 3 | Validation of gene hits in primary culture and analyses of their effect on viral life cycle. a**, **b** Adult (**a**) and weanling (**b**) primary BCECs were transfected with 50 nM siRNA for 72 h against the indicated Group1 hit genes (mainly restriction factors) and LACV infection (LACV sum-intensity-to-Hoechst area ratio) was assessed ($N = 6$ and individual datapoints are shown) Here, $P = 0.0162$, 0.0085, 0.0153, 0.0830, 0.0394, 0.0636, 0.0231 and 0.0271 in (**a**) and $P = 0.6571$, 0.9010, 0.0960, 0.1548, 0.9290, 0.0354, 0.0649 and <0.0001 in (**b**) for si-Bst1, si-Efna2, si-H2q6, si-Clec4e, si-Gja1, si-Ifitm3, si-Ly6c2 and si-Upreg. control vs si-NT, respectively. **c** LACV attachment/ entry assay in bEnd.3 cells transfected with 50 nM of the indicated gene specific and control siRNAs for 72 h and infected with 10 MOI of LACV for 24 h. **d** Representative images of LACV attachment/entry in *Ifitm3* and *Lyc2* siRNA perturbed bEnd.3 cells (green: LACV and blue: Hoechst) ($N = 3$ for **c**, **d**). Here, $P = 0.0682$, 0.0752, 0.9479, 0.3786, 0.3018, 0.0009, 0.0020 and 0.9574 in (**c**, left) and $P = 0.0980$, 0.9622, 0.6042, 0.4154, 0.9939, 0.0132, 0.0011 and 0.0294

in (**c**, right) for si-Bst1, si-Efna2, si-H2q6, si-Clec4e, si-Gja1, si-Ifitm3, si-Ly6c2 and si-Upreg. control vs si-NT, respectively. **e** Plaque assays (each dot = 1 sample) performed at 24 hpi in LACV-infected bEnd.3 cells transfected for 72 h with 50 nM of the indicated gene-specific siRNA. Here, $P = 0.5075$, 0.0030, 0.7566, 0.1347, 0.0120, 0.7941 and 0.0156 for si-Bst1, si-Efna2, si-H2q6, si-Clec4e, si-Gja1, si-Ifitm3 and si-Ly6c2 vs si-NT, respectively. **f** bEnd.3 cells were transfected with 50 nM of the indicated siRNA for 72 h, infected with 10 MOI LACV and confocal microscopy was used to image LACV infection (green), actin (red: phalloidin) and nuclei (blue: Hoechst). **a**, **b** Multiple, two-tailed paired t-tests and **c–e** multiple, two-tailed unpaired t-tests between si-NT and the targeted genes were performed (*$P < 0.05$, **$P < 0.01$, ***$P < 0.001$ and ****$P < 0.0001$) and mean ± SD (3–6 samples per condition) are shown. **d**, **f** Images are representative of 25–75 independent fields. Scale bar = 100 um (white, for regular images), scale bar = 10 um (yellow, for zoomed in images).

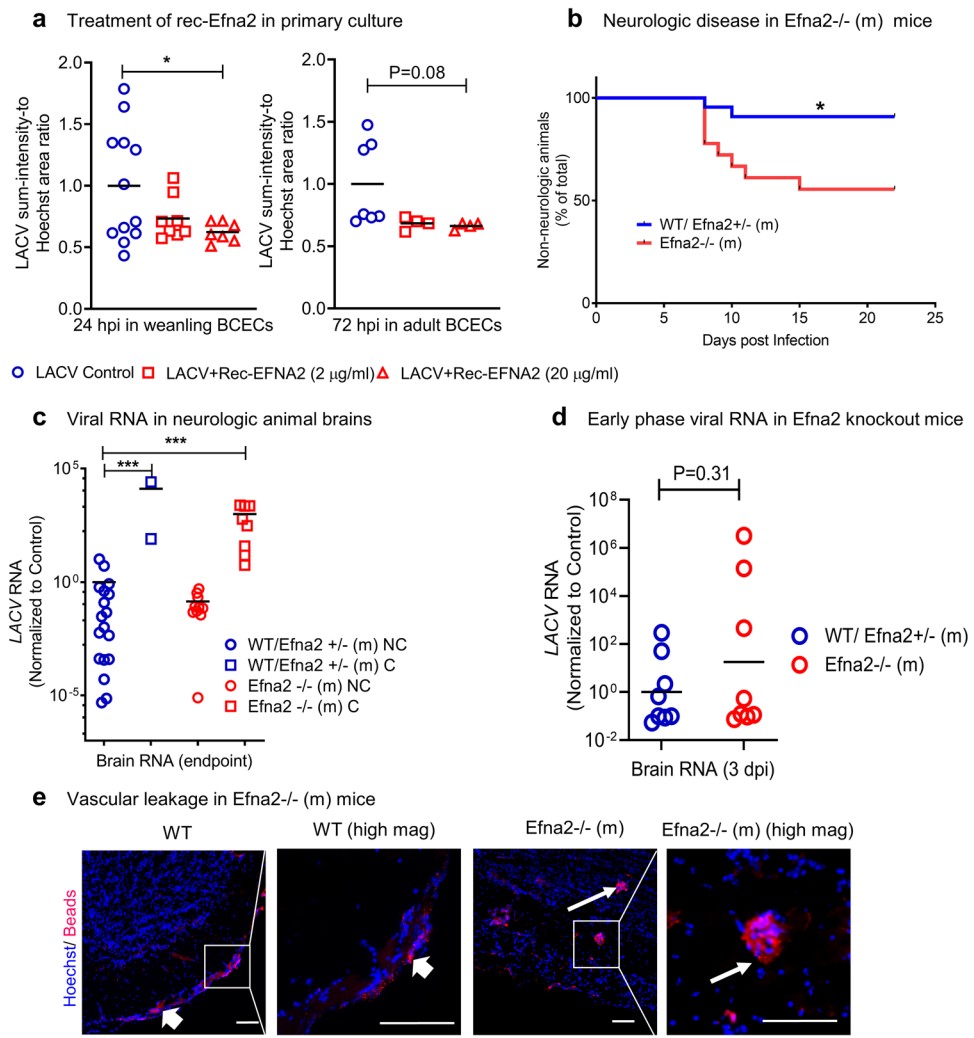

**Fig. 4 | Role of adult-specific higher expression of *Efna2* in protecting BCEC from LACV infection. a** Effect of recombinant EFNA2 (2 ug/ml to 20 ug/ml) on LACV infection levels in weanling and adult BCECs at 24 and 72 hpi (*N* = 4–12 individual samples, data collected based on 25 images for each sample, *P* = 0.1654 and 0.0475 for 2 ug/ml and 20 ug/ml, compared to control, in weanling BCECs and *P* = 0.1068 and 0.0846 for 2 ug/ml and 20 ug/ml, compared to control, in adult BCECs). **b** Adult *Efna2*^−/−^ (m), *Efna2*^+/−^ (m) and WT mice were infected with 10^5 PFU LACV (IP) and the percentage of neurologic mice are shown. See Suppl. Table 3 for mixed *Efna3/5* genotypes of *Efna2*^−/−^ (m) and *Efna2*^+/−^ (m) mice (*N* = 22 WT or *Efna2*^+/−^ (m) mice and *N* = 18 *Efna2*^−/−^ (m) mice, *P* = 0.0105). **c** LACV RNA levels in the brains of infected mice at 8–22 dpi (NC: nonclinical mice and C: clinical mice where *N* = 18 WT or *Efna2*^+/−^ (m) NC mice, *N* = 2 WT or *Efna2*^+/−^ (m) C mice, *N* = 10 *Efna2*^−/−^ (m) NC mice

and *N* = 8 *Efna2*^−/−^ (m) C mice. Here, *P* = 0.0008 for both WT/ *Efna2*^+/−^ (m) C and *Efna2*^−/−^ (m) C, compared to WT/ *Efna2*^+/−^ (m) NC control. **d** Vascular leakage as assessed by the measurement of viral RNA in LACV-infected *Efna2*^−/−^ (m) and WT mice brains at 3 dpi (*N* = 8 mice brains regarding each condition analyzed). **e** Imaging to detect vascular leakage in brain slices from LACV-infected WT and *Efna2*^−/−^ (m) mice at 3 dpi (arrow, FluoSphere beads: red, Hoechst: blue). Thick arrows represent localization of FluoSphere beads at CNS periphery in WT mice whereas the thin arrows represent leakage of FluoSphere beads into the brain parenchyma in *Efna2*^−/−^ (m) mouse (1 out of 4 mice showed leakage, which is represented here). (**a**, one way ANOVA followed by post-hoc Dunnett's test, **b**, Mantel-Cox log rank test and **c**, **d** multiple unpaired, two-tailed t-tests, *P* < 0.05, **P* < 0.01 and ***P* < 0.001).Scale bar = 100 um.

neurologic disease, regardless of their *Efna3* or *Efna5* genotype (Supplementary Table 3). In contrast, most of the WT or *Efna2*^+/−^ (m) mice (~91%) did not show any signs of neurological disease (Fig. 4b). We also observed high levels of virus in the *Efna2*^−/−^ (m) mice with clinical (C) neurological disease versus non-clinical (NC) mice (Fig. 4c). Thus, *Efna2* appears to have an inhibitory role in LACV pathogenesis in vivo (Fig. 4b, c).

To determine if *Efna2* deficiency affects BBB permeability and LACV neuroinvasion, we examined mice at 3 dpi, four days prior to the onset of clinical signs. LACV RNA was detected in approximately ½ of the *Efna2*^−/−^ (m) mice, but in only one mouse from the WT and *Efna2*^+/−^ (m) group (Fig. 4d). These LACV-infected *Efna2*^−/−^ (m) mice were also injected with ~100 nm red FluoSphere beads (virus-sized particle; red staining) at 3 dpi to monitor vascular leakage. Fluorospheres were primarily detected within CNS blood vessels in

WT mice (thick white arrows). However, in one of the four *Efna2*^−/−^ mice, fluorospheres were detected within the brain parenchyma (thin white arrows, Fig. 4e) suggesting early vascular leakage. Thus, the 25–50% increased disease in *Efna2*^−/−^ (m) mice correlated with increased incidence of early viral RNA detection in the CNS and detection of vascular leakage in a subset of these mice. These data indicate that *Efna2* may be an important restriction factor in adult mice to prevent LACV neuroinvasion.

To clarify whether *Efna2* itself was the primary mediator of viral restriction, we compared LACV susceptibility in adult (> 6 weeks old) WT and *Efna2*^−/−^3^+/+^5^+/+^ mice (*Efna2* single knockout (KO); abbreviated as *Efna2*^−/−^ (s)). As expected, the adult WT mice were not susceptible to a 10^5 PFU/mouse LACV dose (~96% non-neurologic). In contrast, approximately 38% of adult *Efna2*^−/−^ (s) mice developed neurological disease in response to LACV infection (Fig. 5a).

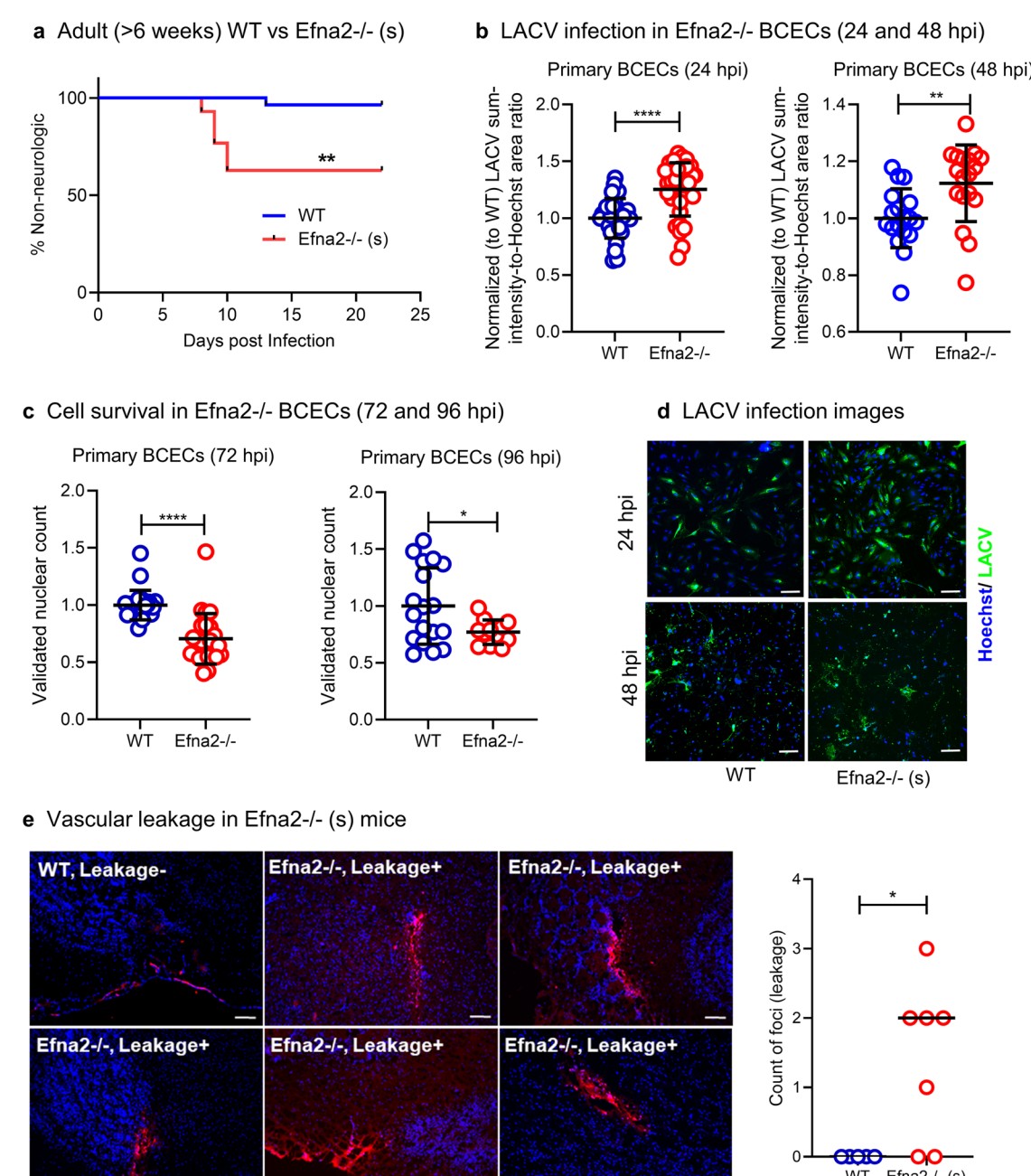

**Fig. 5 | Loss of *Efna2* confers susceptibility in adult LACV-resistant mice.**
**a** Comparison of LACV-induced (10⁵ PFU/mouse dose (IP)) neurologic disease in adult WT and *Efna2⁻/⁻* mice. (*P < 0.05, ****P < 0.0001, Mantel-Cox log rank test, N = 28 WT mice and N = 43 *Efna2⁻/⁻* mice, P = 0.0013). **b** Comparative infection of BCECs after LACV infection (10 MOI) at 24 and 48 hpi on WT and *Efna2⁻/⁻* primary BCECs (N = 33 and N = 18 for WT and *Efna2⁻/⁻* BCECs, mean ± SD is shown, P < 0.0001 and P = 0.0041 for 24 and 48 hpi, respectively). **c** Comparative cell viability (validated nuclear count) of BCECs after LACV infection (10 MOI) at 72 and 96 hpi on WT and *Efna2⁻/⁻* primary BCECs (N = 24 and N = 12-18 for WT and *Efna2⁻/⁻* BCECs, mean ± SD is shown, P < 0.0001 and P = 0.0304 for 72 and 96 hpi, respectively). All individual datapoints are shown and statistical significance analyzed by multiple, two-tailed unpaired t-tests, *P < 0.05, **P < 0.01, and ****P < 0.0001). **d** Representative images at 24 and 48 hpi comparing WT and *Efna2⁻/⁻* BCECs (Hoechst: Blue and LACV: green). Scale bar = 100 um (images are representative of 25 individual images/ sample). **e** One representative WT brain section (without any vascular leakage) is shown along with 5 *Efna2⁻/⁻* brain sections demonstrating vascular leakage (Hoechst: blue and FluoSphere beads: red and scale bar = 100 um) (N = 5 WT brains and N = 7 *Efna2⁻/⁻* brains, of which 5 brains showed probable foci of vascular leakage). The number of foci/ brains was compared using an unpaired t-test (P = 0.0195).

Furthermore, microvessel fragments isolated from *Efna2⁻/⁻* (s) mice had higher levels of LACV infection compared to WT at 24 and 48 hpi (Fig. 5b, d) and reduced survival at 72 and 96 hpi (Fig. 5c). We also examined vascular leakage in vivo using fluorospheres in WT and *Efna2⁻/⁻* (s) mice using a slightly later timepoint of 5 dpi. WT adult mice showed no vascular leakage, while *Efna2⁻/⁻* (s) mice had consistent detection of 1–3 fluorescent bead foci in the brain parenchyma (Fig. 5e, 5 out of 7 mice). Thus, *Efna2⁻/⁻* (s) mice were more susceptible to LACV-induced neurological disease, which correlated with an increase in vascular leakage in the CNS prior to disease onset. This also correlated with increased virus infection and decreased cell survival in *Efna2⁻/⁻* (s) BCECs compared to WT BCECs (Fig. 5b, c). Collectively, these data indicate an important role for *Efna2* in BBB integrity during LACV infection.

## Small molecule-based activation of Cx43 protein delays LACV-induced neurological endpoint in weanling animals

Since knockdown of *Gja1* (which expresses the Cx43 protein) impacted LACV infection in vitro, we hypothesized that activating or upregulating Cx43 might reduce virus infection and pathogenesis. 4-Phenylbutyric acid (4-PBA) is a known activator and channel formation enhancer for several connexins, including Cx43. We infected bEnd.3 cells with LACV +/− 4-PBA and immunostained for LACV and Cx43 at different time points. 4-PBA treatment significantly reduced LACV infection and increased host cell survival at 96 hpi (Fig. 6a). Using confocal microscopy, we observed that the reduction of LACV infection (LACV: green) at 96 hpi correlated with enhancement of Cx43 puncta formation (Cx43: red). The increase in Cx43 puncta formation (white arrows; denoted by granular or laminar Cx43 staining) or localization toward the cell surface was 4-PBA dose dependent (Fig. 6b)[25]. As observed in previous studies[25], the upregulation of Cx43 protein level was confirmed by western blot (Fig. 6c). Thus, 4-PBA-induced upregulation in Cx43 protein levels or puncta formation correlated with inhibited of LACV infection in vitro.

To examine if 4-PBA could inhibit LACV-induced neurological disease, LACV infected weanling mice (~21–23 days old) were treated daily with 500 mg/kg 4-PBA from 0 to 5 dpi (IP injection, LP). LACV infected and vehicle treated mice (LV) were analyzed in parallel. The LV mice exhibited neurological disease between 6 and 8 dpi. In contrast, LP mice had delayed development of neurological disease (Fig. 6d). At the neurological endpoint, virus levels and *Gja1* mRNA expression levels were similar in vehicle and 4-PBA treated groups (Supplementary Fig. 5). However, earlier analysis at 3 dpi, when virus first invades the CNS, demonstrated that LP animals had reduced virus levels, which correlated with an increase in *Gja1* mRNA expression (Fig. 6e). At 4 dpi, brains were isolated from LV and LP mice and co-stained for the endothelial marker ZO1 and LACV. LV mice showed persistent and prominent presence of LACV staining in the OB and OT region of the brain (Fig. 6f; upper panels). In contrast, 4-PBA-treated mice had either no detectable LACV (3 of 4 mice) or much more limited viral staining (1 out of 4 mice, (Fig. 6f; lower panels)) compared to vehicle controls. Higher magnification images showed prominent LACV infection in brain parenchyma in a highly vascularized area in LV mice, whereas LACV infection was restricted primarily to the vascular lumen at 4 dpi in LP mice (Fig. 6f, right side panels). Notably, virus particles were restricted inside the vessels without establishing a sufficient leakage inside brain parenchyma at 4 dpi. Thus, 4-PBA treatment induced upregulation of Cx43 and could partially restrict viral infection and dissemination in the OB/OT region at the early stage of LACV infection.

## Discussion

Age-related susceptibility to LACV infection is dependent, in part, on the ability of LACV to induce vascular leakage and gain access to the CNS[16]. In this study we adopted a systematic, "screening-to-targeting" approach to uncover the differences in gene expression between weanling and adult microvessel fragments (ex vivo)/ BCECs (in vitro) that may contribute to differential LACV susceptibility. The transcriptomic analyses demonstrated that genes could be divided into several age-specific, immune-related or brain-location specific groups. We then used a targeted siRNA library to investigate the role of those factors in controlling viral infection and associated cell death. From the follow-up screen, validation study and mechanistic analyses, we found several genes, mainly restriction factors, which might be important to control LACV induced vascular leakage in adults and could provide therapeutic targets to enhance LACV resistance in children.

A predictable pathway in this model of BCEC leakage would have been alteration of TJ proteins. Several viruses, including hepatitis C virus and Dengue virus, utilize TJs to enter the cell[26] whereas LACV entry factors are largely unknown[21,27]. In our primary screen,

knockdown of the TJs *Cldn2* and *Cldn5* resulted in reduced LACV infection, whereas *Cldn1* knockdown showed no significant effect. However, the influence of these genes was studied in a monolayer culture without sheer-stress. As the conformation of BCECs is very important inside the blood vessels in vivo, the perturbation of TJs might contribute to LACV entry in the susceptible cells. On a related note, we observed that the alteration of GJ expression might be an important factor to control LACV infection and damage to BCECs. We specifically observed that one of the major brain endothelial GJ proteins, Cx43, expressed by *Gja1* mRNA[28] helps controlling LACV infection in bEnd.3 cells.

From the RNA-seq, targeted siRNA screen and other follow up analyses, we studied several genes which are not directly involved in maintaining cell junctions including *Bst1, Clec4e, Efna2, H2q6, Ifitm3* and *Ly6c2*. Although *Bst1* is a marker of endothelial stem cells having regenerative properties[29] and prevalent in weanling BCECs, this gene did not have any major regulatory function in our study. Well characterized immune regulators, like *Clec4e, H2q6* and *Ifitm3*, have defined roles in LACV or other viral infections. *Clec4e* induces an antiviral response against LACV[21] and limited viral infection and plaque production (partially) in our study[30]. *H2q6*, a part of MHC class I, has important functions in antigen presentation[30] and, specifically in endothelial cells, it modulates the actin cytoskeletal network and cell survival[31,32]. LACV infection can lead to actin-network remodeling[16] that may be associated with cytopathic effects observed in cultured BCECs[17]. Interestingly, knockdown of *H2q6* produced higher viral infection and syncytia-like aggregation formation that was associated with fragmentation of F-actin. *Ifitm3* is a known viral restriction factor that blocks entry[33] or pH-dependent fusion[34]. Accordingly, knockdown of *Ifitm3* induced higher LACV infection intensity in BCECs with higher numbers of infected cells. Although, endothelial *Ly6c2* does not have a previously reported role in viral infection, we observed enhancement of viral attachment/entry, infection, replication, and dissemination of LACV with *Ly6c2* knockdown. Although we focused our follow-up analysis on putative restriction factors, *Mmp-25* was the one weanling BCEC enhanced putative susceptibility factor whose knockdown reduced viral infection in primary BCECs, which may warrant further analysis. Altogether, these observations suggest that the age specific LACV susceptibility of BCECs is controlled by a complex multigenic network of factors that facilitate host resistance in adults.

Among these factors, adult enhanced *Efna2* showed a distinctive phenotype in controlling viral infection in bEnd.3 cells and primary BCECs and in viral plaque production. *Efna2* is a signaling molecule that belongs to the Ephrin (Efn) family, which through cognate Eph receptor interactions are involved in angiogenesis and maintenance of endothelial cells. *Efna2* has been recently established to have roles from neuronal development, differentiation and migration[35–37] to angiogenesis and cancer metastasis[23]. In our study, we observed direct treatment of BCECs with rec-EFNA2 reduced viral infection in vitro. The initial observation of increased LACV susceptibility of *Efna2*[−/−] (m) adult mice with mixed *Efna3/5* deficiency suggested that the EphrinA molecules might have a role in resistance to LACV-induced neurologic disease. This was further confirmed by the observed susceptibility of *Efna2*[−/−] (s) single KO mice, suggesting that the *Efna2* gene is a critical factor for LACV resistance in adult mice. Similarly, the BCECs isolated from *Efna2*[−/−] (s) mice were more susceptible to in vitro LACV infection and infection-induced cell death. Also, a significant number of LACV infected *Efna2*[−/−] (s) mice showed leakage of virus-sized particles into the CNS parenchyma. This suggests a pivotal role for *Efna2* in conferring viral resistance. Whether *Efna2* could be targeted in vivo for therapeutic purposes will be an important topic for future investigation.

As noted earlier, CJ proteins are prime targets for regulating viral entry through the BBB and we identified the *Gja1* gene product Cx43 as another potential restriction factor. Previous studies have shown

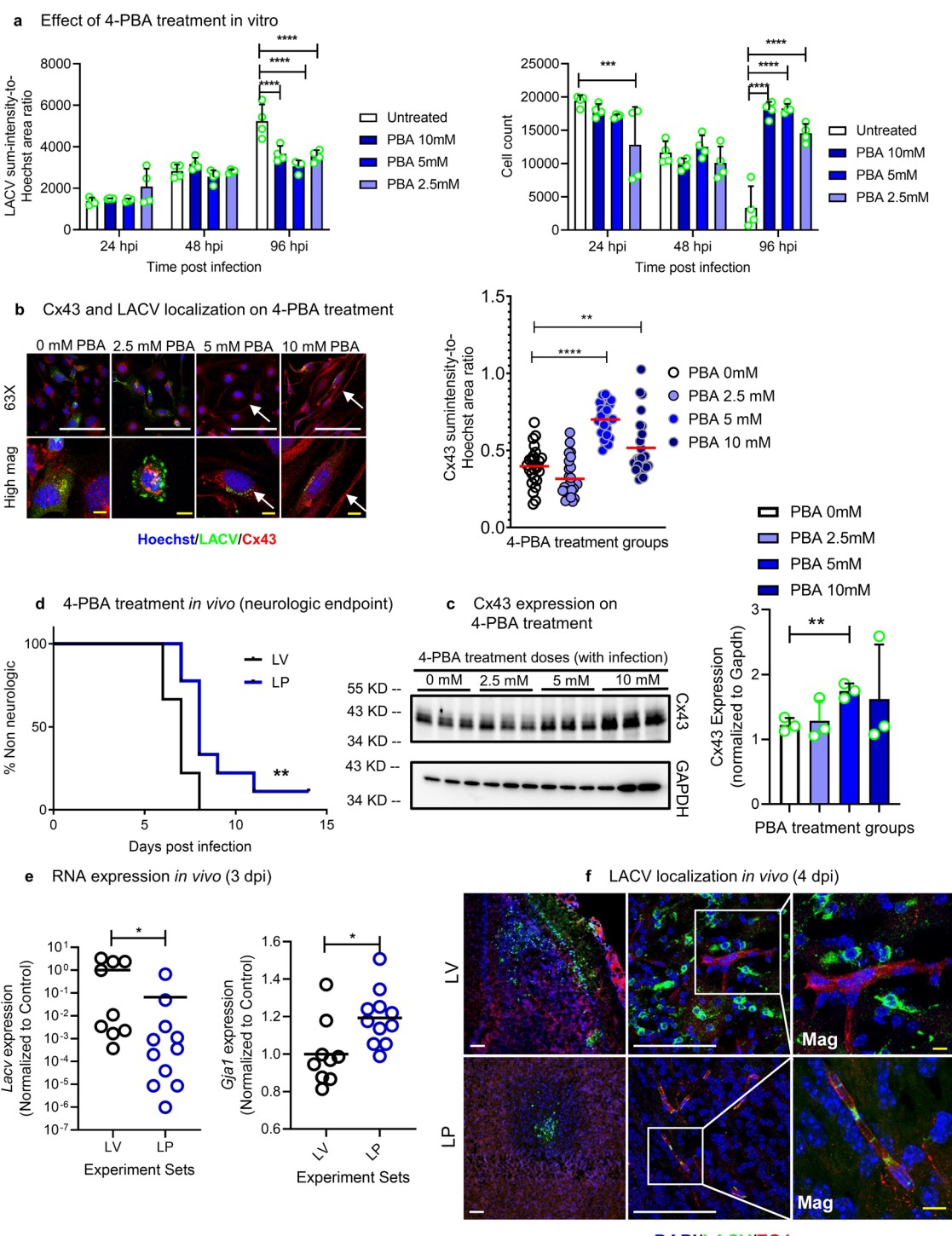

**Fig. 6 | 4-PBA induced alteration of Cx43 (expressed by *Gja1* mRNA) alters LACV induced vascular leakage and viral entry into weanling mice brain. a, b** LACV-infected (10 MOI) bEnd.3 cells, were treated with 4-PBA (0–10 mM) and at different hpi, imaged to measure (**a**) viral intensity or virus-induced cell depletion (*N* = 4 samples, data collected based on 25 images obtained from each sample and mean with SD is shown) or (**b**) localization of Cx43 (red), LACV (green) and Hoechst (blue) (representative of ~12–25 images obtained each condition, *P* = 0.1142, < 0.0001 and 0.0097 from PBA 2.5, 5and 10 mM, compared to PBA 0 mM). White arrows represent punctate staining of Cx43. **c** Western blot and densitometric analyses of Cx43 protein levels in 4-PBA treated bEnd.3 cells. *Gapdh* is a loading control and normalization factor for quantitation (*N* = 3 each condition and mean with SD is shown, *P* = 0.7647, 0.0043 and 0.4589 for PBA 2.5, 5 and 10 mM those are compared to PBA 0 mM). **d** Weanling mice, infected with 2000 PFU of LACV, were

treated with vehicle (LV) or 500 mg/kg-day 4-PBA (LP) until 5 dpi and assessed for neurologic endpoint (*N* = 9 mice for each condition, *P* = 0.0074). **e** Effect of 4-PBA treatment on the correlation between *LACV* and *Gja1* expression in mouse brains at 3 dpi (*N* = 9 LV and *N* = 11 LP brain RNA samples obtained from mice, *P* = 0.0311 and 0.0147 for *LACV* and *Gja1* comparison). **f** Effect of 4-PBA on entry and localization of LACV as assessed by confocal imaging of mouse brain slices at 4 dpi (red: ZO1, green: LACV, blue: Hoechst/nuclei). (**a**) two-way ANOVA followed by Dunnett's test and all groups were compared with vehicle for same time pi, (**b**) one-way ANOVA followed by Dunnett's test, individual points represent separate imaging fields, (**c**) multiple, two-tailed unpaired t-tests, **d** Mantel-Cox log rank test and **e**, two-tailed unpaired t-test where **P* < 0.05, ***P* < 0.01, ****P* < 0.001 and *****P* < 0.0001 and *N* = 3–4 for each experiment. Scale bar = 100 um (white, for regular images), scale bar = 10 um (yellow, for zoomed in images).

viruses like Rous sarcoma virus[38], HIV[39] or mouse hepatitis virus infection[40,41] can modulate Cx43 expression and function in different cell types. Classical swine fever virus infection also depleted Cx43 in endothelial cells[42], but the role of Cx43 and whether its enhancement could limit bunyaviral infection has not been previously investigated. With an aim to reduce viral leakage in susceptible weanling animals, we utilized 4-PBA, a known and well-characterized small molecule activator of Cx43[43]. 4-PBA upregulates a luminal endoplasmic reticulum (ER) protein of 29 kDa (ERp29)[44,45], which controls biosynthesis and trafficking of Cx43[46]. While it has yet to be determined whether the activation of chaperones like ERp29 is the underlying mechanism of 4-PBA induced reduction of LACV in BCECs, this strategy represents a promising approach to combatting LACV-induced encephalitis. It is noteworthy that 4-PBA administration at the onset of viral exposure was sufficient to diminish viral entry and neurological symptoms, which may indicate a particularly rapid response to this potential therapeutic.

In summary, we have utilized a systematic transcriptomic and gene perturbation screening approach to identify genes or gene products involved in controlling LACV susceptibility of BCECs and infection-induced vascular leakage. We identified EFNA2 and Cx43 as two important factors supporting adult-specific resistance to LACV infection. The identification of two different molecular regulators involved in regulation of LACV neuroinvasion suggest synergism between multiple effectors that regulate differences between the immature and mature BBB that impact the ability of viruses such as LACV to gain access to the CNS and cause neuroinvasive disease.

## Methods

### Ethics statement
All animal studies were conducted under animal protocol RML-2018-018-E, LISB 3E, and LISB 4E, adhering to the Principles of Laboratory Animal Care and in accordance and approval by the NIH/NIAID/RML Institutional Animal Care and Use Committee. No human samples were used in this study. The temperature range for both inside the animal cage and the animal holding room was 20–24 °C and the humidity range was 30–70%. A 12 h light/ dark cycle (6 am–6 pm) was maintained at all times. All mice were euthanized after the experiment.

**RNA-sequencing.** Weanling (~3 weeks old) and adult (> 6 weeks old) mice were treated with 100 ug of Poly I:C (HMW) diluted in a 0.5 mg/mL LyoVec transfection reagent solution (Invivogen) via a 100 ul retroorbital (IV) injection. LyoVec reagent was reconstituted using the provided deionized sterile water per the manufacture's recommendations and was used alone as the vehicle control. Microvessel fragments were removed from treated and vehicle animals at 3 h post injection. For comparison of OB versus CT microvessel fragment transcript expression from LACV and mock infection conditions, C57BL/6 weanling mice were infected with a 2000 PFU IP dose of LACV diluted into 200 ul of sterile, pharmaceutical grade PBS. Mock mice received an equivalent volume of uninfected Vero supernatant diluted in PBS. Microvessel fragments from OB and CT were isolated from LACV and mock infected mice at 3 dpi. RNAs were extracted from microvessel fragments as described below and purified using Agencourt RNAClean XP beads (Beckman Coulter, Brea, CA). RNA concentration was measured using RiboGreen fluorescent RNA quantification method (Invitrogen Corporation, Carlsbad, CA). RNA purity and integrity was assessed using Nanodrop 8000 UV spectrophotometry (Thermo Fischer Scientific, Waltham, MA) and the Bioanalyzer RNA 6000 Pico chip assay (Agilent Technologies, Santa Clara, CA), respectively. Sequencing libraries were generated from 115 ng (Adult/Weanling) or 170 ng (OB/CT) RNA using the TruSeq Stranded mRNA sample preparation kit, according to the manufacturer's

recommended protocol (Illumina, Inc., San Diego, CA). Final, purified TruSeq libraries were quantified using the Kapa SYBR FAST Universal qPCR kit for Illumina sequencing (Kapa Biosystems, Wilmington, MA), diluted to 2 nM, and pooled equally. Libraries were sequenced on the HiSeq 2500 instrument using the TruSeq Rapid PE Cluster kit and Rapid 200 cycle SBS kit (Illumina, Inc., San Diego, CA).

**RNA-seq Analysis.** Adapter sequences were removed from raw fastq files using CutAdapt[47] and quality-filtered and trimmed using the FASTX toolkit 0.0.14 (Hannon Lab, CSHL, RRID:SCR_005534). Quality filtered and trimmed reads were mapped to mouse reference sequence GRCm38 using Hisat2 version 2.1.0 with −no-unal −no-mixed parameter settings[48]. The alignment files were used as input to HTSeq-count of the HTSeq version 0.9.1 Python library[49] to generate counts of genes using the GRCm38 reference annotation file. The raw counts matrices were used as input for differential expression analysis using DESeq2[50]. Genes were ranked by DESeq's shrunken log fold change (lfc) and adjusted p value. The acquired data was filtered using statistical significance, base-mean (read count of target genes) and log2 fold change and was fed into R (version 3.6.0), Ingenuity Pathway Analysis (IPA, version 84978992, Qiagen) and SIGNAL (version 2.0) software[51] to understand the pathways and genes involved in age dependent immunostimulatory effects on brain microvessels. By comparing different groups, approximately 50 genes were selected for further analysis in LACV infection.

**Infection of mice with LACV.** All animal studies were conducted under animal protocol RML-2018-018-E, LISB 4E and LISB 3E, adhering to the Principles of Laboratory Animal Care and in accordance and approval by the NIH/NIAID/RML or NIH/NIAID/Bethesda Institutional Animal Care and Use Committee. C57BL/6 (obtained from Jackson Laboratories) or Efna2−/−(m) or Efna2−/−(s) mice were maintained in a breeding colony at RML or Bethesda Campus. LACV human 1978 stock, a kind gift from Dr. Richard Bennett, was used to study LACV pathogenesis in the mice[1], as was diluted to the appropriate dose in 200 μl of sterile, pharmaceutical grade PBS for inoculation. Weanling (aged ~3 weeks) were inoculated IP with a low dose of 2000 PFU/mouse, while adults (aged > 6 weeks) were inoculated IP with 10^5 PFU. Mock inoculated mice were inoculated with 200 ul of PBS with the appropriate amount of cell supernatant from uninfected Vero cells to match the volume for virus dilutions. The whole brains or OB-CT regions were isolated separately from LACV and mock inoculated mice at 3 dpi. Supplementary Table 4 represents abbreviation of most of the experimental groups and other nomenclatures used throughout this manuscript.

**Isolation of microvessel fragments from brain tissue.** Brain microvessels were isolated and grown as previously described[17]. Briefly, after transcardial perfusion with sterile PBS, whole brains from weanling and adult mice were isolated, followed by chopping into small pieces. A 1 h digestion was then performed on the brain pieces using 10 mg/ml of collagenase (CLS2; Worthington Biochemical) in Dulbecco's modified Eagle's medium (DMEM; Sigma) in a 37 °C incubator-shaker. The digested tissues were separated from cellular debris and white matter lipids by centrifugation at 1000 g for 20 min, in presence of 20% bovine serum albumin (BSA) solution prepared in DMEM. A solution of 10 mg/ml of collagenase/dispase (Roche Applied Science) in DMEM was added to the resultant pellet and incubated for 45 min to 1 h at 37 °C. The microvessel fragments obtained from this step were separated on a 33% continuous Percoll gradient that was centrifuged at 1000 g for 10 min. Finally, the microvessel fragments were collected and washed twice in DMEM before using them for ex vivo and in vitro experiments. For the OB and CT comparison, those specific regions were isolated using fine tip forceps and the same protocol as above

was followed to isolate microvessel fragments from these different brain regions.

**Culture of primary BCECs and bEnd.3 cells.** The microvessel fragments were plated on chambered slides or multi-well plates coated with Collagen type IV (Sigma) and human fibronectin (Sigma). DMEM supplemented with 20% fetal bovine serum, 1% penicillin/streptomycin, 1% glutamine, 1 ng/ml of fibroblast growth factor-2 (R&D Systems) and 4 µg/ml of puromycin (Sigma) was used for culturing primary BCECs incubated in a humidified 5% CO2, 95% air atmosphere at 37 °C. The mouse endothelial cell line, bEnd.3 (ATCC, CRL-2299) was similarly cultured using DMEM supplemented with 10% fetal bovine serum, 1% penicillin/streptomycin and 1% glutamine. For in vitro experiments, confluent monolayers were inoculated with different MOIs of LACV (for standardization), and an MOI of 10 was used for experiments. After 1.5 h, fresh medium was added on top, and cells were analyzed at desired time points. Mock inoculated samples were maintained in parallel.

**RNA extraction and real time PCR from cells.** Total RNA was isolated from ex vivo isolated microvessel fragments or cultured BCECs using RNA isolation kits from Zymo Research, using manufacturer's protocol. Total RNA was treated with DNase I (Invitrogen) for 30 min at 37 °C and was purified over RNA cleanup columns (Zymo Research). cDNAs were prepared from cleaned up RNA samples using the iScript reverse transcription kit (Bio-Rad Laboratories) following the manufacturer's instructions. cDNA samples were diluted fivefold in Rnase-free water and those diluted samples were used for gene expression analysis by qPCR using SYBR Green SuperMix with ROX (Bio-Rad Laboratories). For all the analyses, *Gapdh* was used as a housekeeping gene control. The primer sequences used in this study are provided in Supplementary Table 5.

**RNA extraction and real-time PCR from mice brain.** The mice brains, upon surgical removal, were stored at −80 °C until they were thawed with the lysis buffer provided in Rneasy Lipid Tissue Mini Kit (Qiagen). The brains were homogenized using a TissueRuptor (Qiagen) and RNA extraction was done using the manufacture's protocol and using the same kit. The concentration was measured using a NanoDrop 1000 Spectrophotometer (Thermo Scientific) and cDNA was synthesized using an iScript cDNA Synthesis Kit (Bio-Rad Laboratories). Same amount of cDNA was used for real-time PCR using the SYBR Green Assay system (Applied Biosystems) for measuring LACV RNA or TaqMan Real-Time PCR Assays (Thermo Scientific) for detecting *Gja1* using manufacturer's protocol. Appropriate housekeeping gene controls were taken and $C_T$ values were measured using a QuantStudio 6 Flex (Applied Biosysterms) machine or Applied Biosystems ViiA 7 system. QuantStudio Real-Time PCR software was used to extract the data.

**Protein extraction and western blot.** The bEnd.3 cell proteins were extracted at 96 hpi (10 MOI of LACV inoculum dose) using RIPA buffer (Thermo Scientific) with protease inhibitor cocktail (cOmplete Tablets, Mini, EDTA-free, Roche), and phosphatase inhibitor cocktail (PhosSTOP EASY Pack, Roche) and protein concentration was measured using a BCA assay kit (Thermo Scientific). Immunoblotting was done using Cx43 (1:500, polyclonal serum, Sigma) and Gapdh (1:1000, AbCam) was used as a normalization control. Uncropped and unprocessed scans of western blots in included in Supplementary Fig. 6.

**siRNA treatment on cell culture.** The primary BCECs and bEnd.3 cells were grown into confluent monolayers and the siRNA transfections were performed according to the manufacturer's protocol. TransIT-TKO transfection reagent (Mirus Bio.) was used for most of the experiments (2 ul/ well of 96 well plate) along with 50 nM of final

concentrations of siRNA. 100–200 ul of total reaction volume was used per well and incubated in an incubator (humidified 5% CO2, 95% air atmosphere) at 37 °C. After ~72 h post transfection, cells were visually checked for transfection efficiency (by comparing si-Lethal and si-NT) and further experiments were carried on. Except for the control siRNAs, we utilized two independent siRNA sequences for each gene with three replicates each. The siRNAs used in this study are described in Supplementary Table 5.

**FluoSphere injection and imaging of vascular leakage.** WT or *Efna2−/−* (m) mice were inoculated with $10^5$ PFU/mouse dose of LACV (IP). At 3 dpi, these mice were retro-orbitally (intravenous; IV) injected with 100 nm FluoSpheres Carboxylate-Modified Microspherebeads (Thermo Scientific). The mice were euthanized after 30 min. By anesthesia followed by transcardial perfusion. The mice brains were taken, processed similarly for cryosections. The cryosections were counterstained with Hoechst, after rehydrating the slides using PBS. After mounting, a Leica DMI 6000B epifluorescence microscope and LAS-X software was used to capture images.

**Small molecule and recombinant protein treatment in vivo.** 4-PBA (Sigma) and rec-EFNA2 (Sino Biological) were dissolved in regular culture medium for cells treatment. Dimethyl sulfoxide (DMSO; (< 0.5%)) was used when any particulate matter was observed in the highest concentration of 4-PBA (10 mM). Following recommendations from a previous report[52], we used a maximum dosage of 500 mg/kg-day and caution was taken to avoid drug-toxicity induced animal death. The weanling mice (~10 g) were injected with 4-PBA (500 mg/Kg-day) that was dissolved in corn oil (with 200–300 ul DMSO/10 ml) and ~200 ul IP injection was given daily. The mice were weighed daily, and the injection volume was adjusted accordingly. Vehicle mice were similarly injected with the DMSO supplemented corn oil solution in absence of 4-PBA.

**Genotyping and selection of *Efna2* KO mice.** The *Efna2−/−* (m) or *Efna2−/−* (s) mice were kindly donated by Prof. David Feldheim (University of California, Santa Cruz). The mice were inbred and checked for the genotypes by using the following method: Ephrin A2 primers are A2-1: 5′-CCG CTT CCT CGT GCT TTA CGG TAT C −3′, A2-2: 5′-GGG CCG GTT GCA TTC CCA GCG−3′ and A2-3: 5′-GTG AGC GCT GTG GGT GAT GGC GGC−3′ where WT is detected by A2-2 and A2-3: 200 bp and *Efna2* mutant is detected by A2-1 and A2-2: 500 bp. Ephrin A3 primers are A3-1: 5′-GGC TTT TTC TAC AAT CTT TTC TCA − 3′, A3-2: 5′-TCA TGT AGG AGA TAC AGG GC − 3′ and A3-3: 5′-ACC AAA GAA CGG AGC CGG TTG GCG − 3′ where WT is detected by A3-1 and A3-2: 500 bp and *Efna3* mutant is A3-2 and A3-3: 200 bp. Ephrin A5 primers are A5-1: 5′-AGC CCA GAA AGC GAA GGA GCA AAG C −3′, A5-2: 5′-ATT CCA GAG GGG TGA CTA CCA CAT T −3′ and A5-3: 5′-TCC AGC TGT GCA GTT CTC CAA AAC A −3′ where WT is detected by Wild Type A5-2 and A5-3: 400 bp and mutant is detected by A5-1 and A5-3: 500 bp. All these primer details were shared with Transnetyx to develop an automated genotyping assay for *Efna2*, *Efna3* and *Efna5* and the genotypes were obtained from them.

**Immunofluorescence assay on cultured cells.** Immunofluorescence studies were done according to a protocol described previously[17]. For standard immunofluorescence, primary BCECs or bEnd.3 cells were plated on a multi-well plate or chambered glass slide. Samples were fixed using 4% paraformaldehyde (PFA) in PBS, followed by permeabilization of the cells with PBS containing 0.5% Triton X-100. Then the cells were blocked with PBS containing 5% heat-inactivated donkey serum and 0.5% Triton X-100. Following incubation with primary antisera (diluted in blocking solution) for 1 h (RT) or overnight (4 °C), nonspecifically bound antibodies were washed off with blocking serum. Finally, samples were labeled with secondary antisera diluted in

blocking solution, washed with PBS, and incubated with Hoechst stain for 10 min. Samples were mounted using Fluoromount-G (Southern Biotech) and dried for curing at room temperature in dark and stored at 4 °C for long term usage or stored with PBS in a 4 °C refrigerator. The images were acquired using a CX7 microscope (Thermo Scientific) or Leica DMI8000 Inverted confocal microscope (Chicago, IL) microscope. The images and quantification data were acquired and/or processed with Cellomics software (Cx7 machine), LAS X (For Leica confocal and epifluorescence microscope), Fiji (1.52n).ImageJ (1.52a, NIH) or Imaris 8 (Bitplane, Switzerland) or FlowJo 10.8.1 software. Graph pad Prism 8 and 9 and Microsoft Excel (2010) were used.

**Tissue processing and imaging.** Mice were transcardially perfused with PBS and brains were taken. Then, the brains were fixed in 10% neutral buffered formalin (NBF) for 24 hrs, followed by equilibrating the brains in 10% sucrose and then 30% sucrose. Following cryoprotection, the brains were mounted using Tissue Plus O.C.T. compound (Fisher HealthCare), frozen and 10 um sections were obtained using a CM1 950 cryotome (Leica Microsystems). The sections were then stained with LACV antibody (1:500–1:800) or a ZO1 antibody (1:250, Thermo Scientific) using a previously described method[41] with minor modifications. Briefly, frozen tissue sections were washed with ice cold 95% ethanol and then PBS at room temperature (RT) to remove cryomatrix. A hydrophobic marker pen was used to create boundaries around the sections. Slides were then washed with PBS (3 times) and incubated with blocking serum (PBS with 0.5% Triton X-100 and 5% donkey serum) at RT. The sections were incubated overnight at 4 °C with the primary antiserum that was diluted in blocking serum. The sections were washed 3 time with PBS and incubated with secondary antiserum diluted in blocking serum for 2 h at RT. All incubations were done in a humidified chamber. Finally, after 3 PBS washes, the nuclei were counterstained with Hoechst. The images were acquired using a Leica DMI8000 Inverted confocal microscope (Chicago, IL) microscope. The images were processed with ImageJ (1.52a, NIH), Fiji (1.52n, NIH) or Imaris 8(Bitplane, Switzerland) software.

**Plaque assay.** The supernatants from the infected as well as mock inoculated bEnd.3 cultures were collected at different hpi. Plaque assays were performed using a previously described method[17]. In brief, the Vero cell monolayers were inoculated with different dilutions of supernatants in DMEM supplemented with 2% FBS and 1% penicillin/streptomycin. MEM containing 1.5% carboxymethyl cellulose (CMC) was added after 1 h and incubated at 37 °C in a humidified 5% $CO_2$, 95% air atmosphere. On 5 dpi, the cells were fixed using 10% formalin solution, washed and the plaques were visualized using 0.35% crystal violet solution.

**Cell viability assay.** The bEnd.3 cells were taken at specific hpi and cell viability was measured using CellTiter-Glo (Promega) reagent, using the manufacturer's protocol. The luminescence was measured using FLUOstar Omega (BMG Labtech). Alternatively, the cell count was measured from the number of nuclei (stained by Hoechst) using a Cx7 imager (Thermo Scientific).

**Attachment/ entry assay.** The confluent monolayers of bEnd.3 cells were prechilled in 4 °C and the cells were infected with 10 MOI of LACV inoculum on ice. Cells were kept in 4 °C for 2–3 h to allow the viral particles only to attach on cell surface but not replicate at the reduced temperature. The inoculum was removed with caution and cells were washed thoroughly (2–3 times) at 4 °C. Fresh culture medium was then added on top and cells were brought to 37 °C. The attached virus was then allowed to amplify in cells at the increased temperature until 24 hpi and were analyzed for LACV intensity using immunofluorescence assay.

**Statistical analyses.** Mean ± SD or individual datapoints with mean are shown in each experiment. For RNA-seq analyses, a negative binomial generalized model with Wald test for significance was performed. The two-sided $p$-values were corrected for multiple testing using the Benjamini Hochberg method. One-way ANOVA followed by Tukey's multiple comparison test or Dunnett's multiple comparison tests are performed as a post-hoc analyses for multiple mean comparison or control mean comparison. Multiple paired or unpaired t-tests are performed in two-group comparisons or screening analyses. Survival comparisons were done using Log-rank (Mantel-Cox) test. The details of the statistical analyses are provided in individual figure legends.

**Reporting summary**
Further information on research design is available in the Nature Portfolio Reporting Summary linked to this article.

## Data availability

The RNA-seq data were deposited to the Gene Expression Omnibus (GEO) and are publicly available (GSE217434). Hyperlink to the dataset: https://www.ncbi.nlm.nih.gov/geo/query/acc.cgi?acc=Gse217434. The SIGNAL software is available to the public using this hyperlink: https://signal.niaid.nih.gov/. The wildtype mice (RRID: IMSR JAX:000664 and Strain #:000664) are available in The Jackson Laboratory. The Efna2$^{-/-}$ mice are available upon request in the laboratory of Dr. Iain Fraser, NIAID. The other datasets used and/or analyzed during the current study are available from the corresponding author upon request. Source data are provided with this paper.

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

## Acknowledgements

We acknowledge Prof. David Feldheim (University of California at Santa Cruz) for kindly donating *Efna2*$^{-/-}$ (m) and *Efna2*$^{-/-}$ (s) mice. We thank Samuel Katz, Jing Sun and Sharat Vayttaden for their kind assistance and training for the transcriptomic screen, siRNA targeted screen and high throughput imaging, respectively. We thank the Rocky Mountain Laboratories Genomics Unit of the NIAID Research Technologies Branch for kind assistance with RNA-seq and follow-up analyses. We specifically thank Stacy Ricklef for RNA-seq library preparation and sequence processing. We also thank the Imaging Core of the NIAID Research Technologies Branch for providing imaging facilities. We acknowledge and appreciate the NIAID 14BS animal caretakers and facility managers for their assistance in maintenance and breeding of the animals. We also thank James Striebel, Simote Foliaki, Sinu John, Clinton Bradfield, and Julia Gross for critically reading the manuscript and the members of the

Laboratory of Persistent Viral Diseases (LPVD), RML and the Laboratory of Immune System Biology (LISB), Bethesda for critical input and assistance during the course of this work. This study was supported by the Intramural Research Program of the National Institute of Allergy and Infectious Disease (NIAID), National Institutes of Health (NIH). R.B. was supported by the RML-Bethesda NIAID Fellowship.

## Author contributions

R.B. performed the experiments and wrote the manuscript. S.G. assisted with the acquisition, processing, and analysis of confocal images. C.W. performed all experiments pertaining to the RNA-seq analysis. S.L.A. and C.M. performed, analyzed, and inventoried the RNA-seq data, and assisted with transcriptomic screen data analyses (IPA analysis, preparing volcano plots, etc.). KEP and IDCF designed the project, oversaw the experiments, and wrote the manuscript. K.E.P. and I.D.C.F. contributed equally. All authors read and approved the final manuscript.

## Funding

## Competing interests

The authors declare no competing interests.

## Consent for publication

All authors have approved the submission of this manuscript.
