## [Peer Review File · Nature Communications]

Identification of age-specific gene regulators of La Crosse virus neuroinvasion and pathogenesisREVIEWER COMMENTS

Reviewer #1 (Remarks to the Author):

Review of Basu 2022

This is a well-written and generally enjoyable manuscript from two labs at NIH / NIAID in Maryland and Montana. The question being asked is straight-forward – are there particular genes that are expressed in juvenile that make them more susceptible to severe viral encephalitis compared with adults. Several candidate genes are identified in initial screens and then followed up with targeted studies. The results show that connexin43 and Ephrin A2 are important genes for severe encephalitis following infection with La Crosse Virus in children. This is a very intriguing study showing potential mechanisms for why this disease can be so devastating in children but less so in adult infection.

While probably necessary due to page restrictions, the abbreviations become confusing at times, and get in the way of initial understanding (especially of groups), at least to this reviewer.

It is interesting that genes described as restriction factors are intimately associated with blood-brain barrier (BBB) function, including connexin43. Combined with VEGFA and altered claudin genes, these are pretty convincing evidence of mechanisms of encephalitis.

It is not immediately apparent why efna2, 3, & 5 mixed genotypes were studied (Fig 4 and S table 2).

Fig4E confuses this reviewer: the Y axis is labeled as ratio to LV, but one of the groups is LV.

Ideally, the high mag images should have what the magnification is for these (eg. Fig 6F)

Attachment / entry assay: there is a question if viruses will actually enter cells at 40C. This has been examined with different viruses over the years, including polio. This reviewer is not aware of LACV actually entering the cells at 40C. Has this been determined. The simple fix here might be to rephrase along the lines of, " ...for 2-3 hours to allow the viral particles only to attach on the cell surface, but not replicate...."

Reviewer #2 (Remarks to the Author):

In this study, Basu and colleagues provide an exciting and timely study of molecular factors driving differential susceptibility to LACV by age. LACV is a medically important virus which has scarcely been studied with an adequate level of sophisticated approaches (due to biocontainment issues, etc) – so the present study is a significant advance. The transcriptomic datasets describing different endothelial factors expressed across age and infection status is a rich resource for the field and unlike any currently existing dataset that this reviewer has seen. The followup siRNA screens identifying important phenotype-driving factors is well designed and resulted in identification of the two major pathways the authors focus on in downstream mechanistic experiments. Overall, this study is well written, well controlled, and addresses important and urgent questions, with implications for LACV in particular, along with broader insights into molecular mechanisms of viral encephalitis more generally.

Figure 1: While an experimental diagram and summary table (Table S1) is provided for the RNAseq datasets, the analysis of the transcriptomic datasets is somewhat shallow. The authors indicate that DEGs with at least a log2 fold change and $Pecam1/CD31	Platelet/endothelial cell adhesion molecule 1	BCECs	16101
Vcam1	Vascular cell adhesion molecule 1	BCECs	20747
Vwf	Von Willebrand factor	BCECs	20977
Gfap	Glial fibrillary acidic protein	Astrocytes	102
Tubb3	Tubulin, beta 3 class III	Neurons	45
Cspg4	Chondroitin sulfate proteoglycan 4	OPCs, pericytes	1419
Pdgfb	Platelet derived growth factor, B polypeptide	Pericytes	4772
Cnn1	Calponin 1	Smooth muscle cells	614

References

1. Blakqori, G., et al., *La Crosse bunyavirus nonstructural protein NSs serves to suppress the type I interferon system of mammalian hosts*. J Virol, 2007. **81**(10): p. 4991-9.
2. Mukherjee, P., et al., *Activation of the innate signaling molecule MAVS by bunyavirus infection upregulates the adaptor protein SARM1, leading to neuronal death*. Immunity, 2013. **38**(4): p. 705-16.
3. Dawes, B.E., et al., *Human neural stem cell-derived neuron/astrocyte co-cultures respond to La Crosse virus infection with proinflammatory cytokines and chemokines*. J Neuroinflammation, 2018. **15**(1): p. 315.
4. Basu, R., et al., *Age influences susceptibility of brain capillary endothelial cells to La Crosse virus infection and cell death*. J Neuroinflammation, 2021. **18**(1): p. 125.
5. Perriere, N., et al., *Puromycin-based purification of rat brain capillary endothelial cell cultures. Effect on the expression of blood-brain barrier-specific properties*. J Neurochem, 2005. **93**(2): p. 279-89.
6. Deli, M.A., et al., *PrP fragment 106-126 is toxic to cerebral endothelial cells expressing PrP(C)*. Neuroreport, 2000. **11**(17): p. 3931-6.
7. Winkler, C.W., et al., *Hyaluronan anchored to activated CD44 on central nervous system vascular endothelial cells promotes lymphocyte extravasation in experimental autoimmune encephalomyelitis*. J Biol Chem, 2012. **287**(40): p. 33237-51.
8. Rosas-Hernandez, H., et al., *Isolation and Culture of Brain Microvascular Endothelial Cells for In Vitro Blood-Brain Barrier Studies*. Methods Mol Biol, 2018. **1727**: p. 315-331.

REVIEWERS' COMMENTS

Reviewer #2 (Remarks to the Author):

The authors have done an excellent job revising this exciting manuscript. I have no further concerns.

Reviewer #3 (Remarks to the Author):

The reviewer commend the authors on their efforts in revising their manuscript. The reviewer had no further comments or issues.